# Exploration of Machine Learning for Hyperuricemia Prediction Models Based on Basic Health Checkup Tests

**DOI:** 10.3390/jcm8020172

**Published:** 2019-02-02

**Authors:** Sangwoo Lee, Eun Kyung Choe, Boram Park

**Affiliations:** 1Network Division, Samsung Electronics, Suwon 16677, Korea; lsw00kor@hotmail.com; 2Department of Surgery, Seoul National University Hospital Healthcare System Gangnam Center, Seoul 06236, Korea; 3Department of Surgery, Seoul National University College of Medicine, Seoul 03080, Korea; 4Department of Biomedical Science, Seoul National University Graduate School, Seoul 03081, Korea; 90may11@gmail.com

**Keywords:** machine learning, prediction, uric acid

## Abstract

Background: Machine learning (ML) is a promising methodology for classification and prediction applications in healthcare. However, this method has not been practically established for clinical data. Hyperuricemia is a biomarker of various chronic diseases. We aimed to predict uric acid status from basic healthcare checkup test results using several ML algorithms and to evaluate the performance. Methods: We designed a prediction model for hyperuricemia using a comprehensive health checkup database designed by the classification of ML algorithms, such as discrimination analysis, K-nearest neighbor, naïve Bayes (NBC), support vector machine, decision tree, and random forest classification (RFC). The performance of each algorithm was evaluated and compared with the performance of a conventional logistic regression (CLR) algorithm by receiver operating characteristic curve analysis. Results: Of the 38,001 participants, 7705 were hyperuricemic. For the maximum sensitivity criterion, NBC showed the highest sensitivity (0.73), and RFC showed the second highest (0.66); for the maximum balanced classification rate (BCR) criterion, RFC showed the highest BCR (0.68), and NBC showed the second highest (0.66) among the various ML algorithms for predicting uric acid status. In a comparison to the performance of NBC (area under the curve (AUC) = 0.669, 95% confidence intervals (CI) = 0.669–0.675) and RFC (AUC = 0.775, 95% CI 0.770–0.780) with a CLR algorithm (AUC = 0.568, 95% CI = 0.563–0.571), NBC and RFC showed significantly better performance (*p* < 0.001). Conclusions: The ML model was superior to the CLR model for the prediction of hyperuricemia. Future studies are needed to determine the best-performing ML algorithms based on data set characteristics. We believe that this study will be informative for studies using ML tools in clinical research.

## 1. Background

In 2016, the deep-mining, computer-programmed Go player alphaGo beat one of the best Go players, Lee Sedol, by a score of 4:1 [1]. While artificial intelligence, machine learning (ML) and deep learning have been increasingly applied in various areas of society, they have not been applied earnestly in clinical research. Most studies using clinical data to date have been analyzed with conventional statistical models. ML algorithms have notable advantages over conventional statistical models. First, ML does not require a specific hypothesis to explain the association between multiple predictors and dependent outcomes. Therefore, unknown significance data that are not expected to be important can be considered in the analysis instead of being overlooked [2]. Second, clinical input factors have complex interactions, and therefore, they are not completely independent. Conventional statistical models have limitations in terms of integrating these issues. In contrast, ML can consider all possible interactions between various input data [3].

In the era of clinical big data, ML methodologies may uncover new information from electronic medical record (EMR)-based clinical big data that has not been discovered by conventional research methods, and thus, they may contribute to medical developments. In this study, we introduced an ML analysis method that has not been applied to clinical data for clinical big data analysis to lay the foundation for clinical data research in the era of artificial intelligence. Hyperuricemia is considered to be a risk factor for the development of metabolic, renal, and cardiovascular diseases [4], and its prediction could be helpful in preventing various chronic diseases. We used several ML tools and algorithms to predict uric acid status based on basic healthcare checkup test results.

## 2. Methods

### 2.1. Data Acquisition

This study aimed to include Korean men and women over 40 years of age who had received self-paid, comprehensive health checkups at Gangnam Center, Seoul National University Hospital, from January 2005 to December 2015. We retrospectively collected data. The uric acid level was measured during comprehensive health checkups at Gangnam Center in addition to tests performed in national checkups. In Korea, the National Health Insurance Corporation (NHIC) funds basic health checkup examination fees annually or biannually. The checkups include blood tests (white blood cell count (WBC), hemoglobin, fasting glucose level, total cholesterol, glutamic oxaloacetic transaminase (GOT), glutamic pyruvic transaminase (GPT), gamma-glutamyl transferase (GGT), creatinine, triglyceride, high-density lipoprotein (HDL) cholesterol, low-density lipoprotein (LDL) cholesterol, urine albumin, anthropometric measurements (blood pressure, height, weight, body mass index, and waist circumference), and self-recorded questionnaires (past medical history of diabetes, dyslipidemia, and hypertension; alcohol intake and smoking). It does not include uric acid levels.

Hyperuricemia is commonly defined in clinical practice as a serum uric acid level above 7.0 mg/dL for men and above 6.0 mg/dL for women [4]. In our interviews, we asked participants about their history of being diagnosed with diabetes/hypertension/dyslipidemia and whether they were currently taking medications for these conditions. Smoking status was subdivided into none, ex-smokers, and current smokers. Alcohol consumption was defined as no (alcohol consumption ≤20 g/day) and yes (consumption >20 g/day).

### 2.2. Study Design

After the collection of the data set, we performed the study in the steps described below.

Step 1. When an observation was missing any categorical predictors, we removed the observation from the data set.

Step 2. When an observation was missing any continuous predictors, we replaced these items with the mean value of the non-missing observations for the predictor.

Step 3. For each of the algorithms outlined in Table 1, we generated a set of parameters for the purpose of parameter tuning in the algorithm.

Step 4. For each of the parameter sets generated in step 3, we trained the algorithm on the training set and calculated performance measures, such as accuracy, sensitivity, specificity, prediction BCR, and F1-score, on the test set (Table 2) [5,6,7].

In particular, we adopted K-fold cross validation with K equal to five; The total data were divided into a training-validation set (ratio of 0.7) and a test set (ratio of 0.3). The training-data set was subdivided into K non-overlapping folds, K-1 folds of which were used for training, and one remaining fold was used for validation. We obtained the K-fold cross-validated results by allowing each of the K folds to be used as the validation fold and averaging the obtained validation results about K.

Step 5. Now, given that the number of K-fold cross-validated results was the same as the parameter set size, we chose some of the parameter sets according to the criteria below.

Step 5-1. We chose a parameter set that maximizes the sensitivity.

Step 5-2. We chose a parameter set that maximizes the specificity.

Step 5-3. We chose a parameter set that maximizes BCR (balanced classification rate) as a metric that considers both sensitivity and specificity simultaneously.

Step 5-4. Likewise, we chose a parameter set that maximizes F-score (F-score: harmonic mean of recall and precision) with beta equal to one.

Step 6. Once we found a parameter set by maximizing the performance measures in step 5, we trained the algorithm again on the training set, but this time, we trained the algorithm on all of the K folds instead of only the K-1 folds.

Step 7. Now, the trained model from step six was used to evaluate the performance measures on the test set.

Step 8. We evaluated the performance of the developed models and compared the best-performing algorithms of each ML model with conventional logistic regression (CLR) by measuring the area under the receiver operating characteristic (ROC) curves (AUC) on the combined set, which includes the training and test sets. Comparisons of the ROC curves were done with the DeLong test [8].

### 2.3. Evaluated Machine Learning Models

We evaluated the most common ML models, namely, discriminant analysis classification (DAC) [9], decision tree classification (DTC) [10], K-nearest neighbor classification (KNNC) [11,12], naïve Bayes classification (NBC) [13,14], random forest classification (RFC) [15,16], and support vector machine classification (SVMC) [17,18].

### 2.4. Tools for Machine Learning and Statistical Analysis

ML analysis was performed using MATLAB 2016B (MathWorks Inc., Natick, MA, USA). In the analyses using conventional statistics, a chi-squared test or ANOVA was used for categorical variables, and a Student’s *t*-test was used for continuous variables. CLR analysis was performed in the prediction model design. In the performance comparison between conventional statistics and ML, a ROC curve was used by calculating its AUC. Conventional statistics were conducted with R 3.2.2 (R Development Core Team; R Foundation for Statistical Computing, Vienna, Austria).

### 2.5. Ethics Statement

The Institutional Review Board of Seoul National University Hospital approved the study protocol (IRB number 1706-058-859), and the study was conducted in accordance with the Declaration of Helsinki. Informed consent was waived by the Board.

## 3. Results

### 3.1. Baseline Characteristics

Data from a total of 55,227 persons were collected during health checkups. Cases with any missing categorical predictors were removed, and data from a total of 38,001 people were analyzed in this study. The number of people who met the definition of hyperuricemia was 7705 (25.4%). The demographic features and characteristics of the population are shown in Table 3.

### 3.2. Performance of the Respective Machine Learning Algorithms in Test Set Population

We performed six ML methods, namely, DAC, DTC, KNNC, NBC, RFC, and SVMC. For the algorithms shown in Table 1, the obtained results are shown in Appendix A.

### 3.3. Overall Comparison of Respective Machine Learning Models

Table 4 summarizes the performance of the different predictive models in the training and test sets for the maximum sensitivity criterion and the maximum BCR criterion. With our K-fold cross validation at K equal to five, severe degradation was not encountered due to overfitting. Since our model is intended to identify the population at risk for hyperuricemia, we developed a model that had a maximum sensitivity criterion as the primary target. For the maximum sensitivity criterion model, the best performance was obtained by the NBC algorithm, which had an accuracy of 0.63, sensitivity of 0.73, and specificity of 0.63 on the test set, and the second-best performance was by the RFC algorithm. However, considering the imbalance of the uric acid data sets, we also aimed for a maximum BCR criterion, and the RFC algorithm showed the best performance with an accuracy of 0.70, sensitivity of 0.64, and specificity of 0.71 on the test set, and NBC had the second-best performance.

### 3.4. Performance Comparison with Conventional Logistic Regression Model

We compared the best-performing algorithms for the maximum sensitivity criterion, NBC and RFC, with the CLR algorithm using the AUC.

The results are shown in Table 5. The NBC (AUC = 0.669, 95% CI = 0.669–0.675) and RFC (AUC = 0.775, 95% CI 0.770–0.780) models showed better performance than the CLR model (AUC = 0.568, 95% CI = 0.563–0.571) with statistical significance (*p* < 0.001).

## 4. Discussion

In this paper, we compared various ML algorithms, namely, DAC, KNNC, NBC, SVMC, DTC, and RFC, for the prediction of hyperuricemia using basic health checkup data. We found that NBC achieved the best performance and that RFC had the second-best performance in terms of sensitivity on the test set. For BCR, on the other hand, the RFC algorithm performed the best and NBC was the second best on the training set. When we compared the performance of ML algorithms and CLR analysis, ML algorithms had higher prediction power, as determined by AUC [8]. A large set of EMR-based clinical data can be used for the prediction of various healthcare issues by ML analysis.

In recent years, ML, artificial intelligence and deep learning have been increasingly used in various fields [19,20,21]. However, there have not been many reports on the application of these methods for disease prediction models using clinical data in the medical field [22]. There are several reasons to choose ML algorithms over conventional statistical method for designing a prediction model. First, compared to conventional statistical analysis, ML can design a prediction model that reflects the relationship between variables without prior knowledge of the algorithm [23]. This characteristic makes it possible to include all information from the input data regardless of its effectiveness during analysis and prevents overseeing data with indefinite effectiveness. Second, in conventional statistical analysis, it is assumed that the input variables are independent [3]. However, this assumption is impossible in the real world. Various input factors are inter-related in complex ways, regardless of whether these ways are known or not. ML considers potential interactions so that all information in the input data can be reflected in the analysis [24], and it can improve prediction performance with complex, heterogenous, and high-dimensional data [25].

In this study, hyperuricemia was targeted as one of the tasks used to create a disease prediction model using ML based on basic clinical information. We have chosen the disease entity “hyperuricemia” as the output of the prediction model because hyperuricemia is known to be related to various chronic diseases [4]. Thus, hyperuricemia can be a biomarker of various chronic diseases and reflects one’s health status. However, uric acid levels are not routinely measured at basic health checkups. If we use the prediction model designed by the ML method to screen someone at high risk of hyperuricemia, we could recommend a uric acid level test to individuals who need an examination. This approach could represent the beginning of precision medicine with respect to health checkup tests.

At our institute, visitors perform self-paid comprehensive health checkup tests, which include expensive, advanced tests. In Korea, the NHIC pays each participant’s basic health examination fee once every two years for people aged 40 years or older. The test items included in this study were used as input factors, and the uric acid level, which is a test that is not included in the basic examination, was set as an output factor.

In this study, a prediction model was designed that included not only well-known risk factors of hyperuricemia, such as aging, obesity, high alcohol intake, hypertension, and cholesterol level [26,27,28], but also factors with no clear relation to the disease. These factors would have been removed in CLR. However, in the case of ML, we designed the prediction model by including all factors with a marginal effect and factors with unknown associations. In the hyperuricemia prediction model, NBC (1st) and RFC (2nd) showed the best performance in the test set for maximizing the sensitivity criterion. KNNC showed high sensitivity in the training set (sensitivity = 1), but this performance was not validated in the test set (sensitivity = 0.34). We selected the criterion model that maximized sensitivity because the role of these models was to assign red flags to individuals with an unexpectedly high risk of hyperuricemia and recommend further evaluation based on basic test results. At the same time, we also evaluated the model that maximized BCR. BCR is an average of specificity and sensitivity; BCR can only be high when both sensitivity and specificity are high [7]. Therefore, it can reflect performance in terms of both sensitivity and specificity. This metric provides a more precise measure of the effectiveness of the classifier than other metrics [29]. Therefore, we also evaluated models that maximized BCR in order to consider the balance between sensitivity and specificity. In the BCR maximizing model, RFC (1st) and NBC (2nd) showed the best performance, which is somewhat consistent with the results with models maximizing sensitivity.

In terms of the characteristics and advantages of particular ML algorithms [30], NBC is a probabilistic model that uses the naïve independence assumption [31] and can analyze uncertain medical data [32,33,34]. The main advantage of NBC is that it takes into account all available information to design the model [30,35,36]. It is known to be a useful classifier for clinical decision support [20] and has been used in several medical data analyses [30].

Random forest is a ML technique and is an ensemble learning method used for classification or regression from many decision trees [16,37]. In this paper, we employed Breiman’s random forest algorithm by using Matlab’s treebagger function [15,38]. RFC is used in medical studies, such as proteomics and genetics studies [39,40,41], but it is not actively applied to clinical data. There are several advantages of RFC, and the most crucial ones for our prediction design were that (a) it has a relatively lower risk of overfitting and that (b) it can include continuous and categorical variables in the analysis [42]. Our data set had the following characteristics: (1) It includes both categorical and continuous input factors, (2) the predictive power of the input factor is not well known or has borderline power, (3) the purpose of the model is general application for public health checkups, so overfitting should be avoided, and (4) the input factors are numerous with high dimensionality. Based on these characteristics, it is reasonable that NBC and RFC are the models that showed the highest performance. Consequently, an appropriate ML tool should be selected based on the characteristics of the input data and the purpose of the prediction design.

The development of an algorithm for predicting high-dimensional clinical information through ML using EMR-based basic clinical information may have the following benefits. First, it may play a role as a supervision tool for selecting undetected and unsuspected high-risk populations using limited information. Second, by introducing a ML tool that has not yet been actively applied in medical clinical studies into the medical big data analysis, EMR-based medical big data that have already accumulated can produce information that leads to new clinical knowledge. Third, this prediction model can save medical expenses by selecting patient groups that need to be closely examined and recommending certain tests. This model is also expected to be a basic tool to promote health by highlighting which people need tests and conducting additional screenings. Hyperuricemia is known as a predicting factor for the development of various chronic diseases. By inputting the basic laboratory test results in this model, we could identify those who need special medical attention among antecedently known healthy populations.

Our study has several limitations. First, our study was performed in a population who participated in an expensive, self-paid health checkup program. The effect of socioeconomical status may limit the generalization of our results to other populations. Second, it is difficult to interpret the results of an ML model. Compared to conventional statistics, which assess the effect of individual predictors, the process and effect of each predictor are not visualized in ML.

## 5. Conclusions

In this study, a large clinical set was used to develop a prediction model for high-risk health status by applying various ML tools and evaluating their performance. The best ML model was superior to a conventional model developed by a CLR model as per estimates by AUC. Future studies are needed to determine the best-performing ML algorithms based on the characteristics of the data set. We believe that this study will be informative for studies using ML tools in clinical research.

## Figures and Tables

**Table 1 jcm-08-00172-t001:** Compared machine learning algorithms.

No.	Machine Learning Scheme	Method in Detail	Data Splitting Method
1	Discrimination analysis classification (DAC)	K-fold cross validation with k = 5	Training set ratio = 0.7, test set ratio = 0.3
2	k-nearest neighbor classification (KNNC)	K-fold cross validation with k = 5	Training set ratio = 0.7, test set ratio = 0.3
3	Naïve Bayes classification (NBC)	K-fold cross validation with k = 5	Training set ratio = 0.7, test set ratio = 0.3
4	Support vector machine classification (SVMC)	K-fold cross validation with k = 5	Training set ratio = 0.7, test set ratio = 0.3
5	Decision tree classification (DTC)	K-fold cross validation with k = 5	Training set ratio = 0.7, test set ratio = 0.3
6	Random forest classification (RFC)	K-fold cross validation with k = 5	Training set ratio = 0.7, test set ratio = 0.3

**Table 2 jcm-08-00172-t002:** Performance measures and their definitions.

Notation	Description	Upper Bound
Accuracy	(TP + TN)/(TP + FN + FP + TN)	1 when FN = 0 and FP = 0
Sensitivity (Recall, True positive rate)	TP/(TP + FN)	1 when FN = 0
Specificity (True negative rate)	TN/(FP + TN)	1 when FP = 0
Precision	TP/(TP + FP)	1 when FP = 0
Balanced classification rate	(SN × SP)^1/2^	1 when SN = 1 and SP = 1
F1-score	(2 × SN × Precision)/(SN + Precision)	1 when SN = 1 and Precision = 1

TP: true positive; TN: true negative; FP: false positive; FN: false negative; SN: sensitivity; and SP: specificity.

**Table 3 jcm-08-00172-t003:** Demographics features of the included population.

	Normal Uric Acid Level (*N* = 30,296)	Hyperuricemia (*N* = 7705)	*p*
Sex (*N*, %)			
Male	19,540 (64.5%)	6764 (87.8%)	<0.001
Female	10,756 (35.5%)	941 (12.2%)	
Age	52.1 ± 9.4	50.7 ± 9.6	<0.001
Systolic blood pressure	116.6 ± 13.9	120.0 ± 13.3	<0.001
Diastolic blood pressure	75.6 ± 10.8	79.2 ± 10.7	<0.001
Height (cm)	166.2 ± 8.0	169.3 ± 7.1	<0.001
Weight (kg)	65.2 ± 11.0	72.4 ± 10.9	<0.001
Body mass index (m^2^/kg)	23.5 ± 2.8	25.2 ± 2.9	<0.001
Waist circumference	84.7 ± 7.9	89.4 ± 7.7	<0.001
White blood cell count (cells/mL)	5.4 ± 1.5	5.9 ± 1.7	<0.001
Hemoglobin (g/dL)	14.4 ± 1.5	15.1 ± 1.3	<0.001
Glucose (mg/dL)	97.6 ± 19.5	99.0 ± 18.2	<0.001
Total cholesterol (mg/dL)	193.1 ± 34.2	200.8 ± 36.0	<0.001
GOT (IU/L)	24.4 ± 14.8	28.5 ± 16.7	<0.001
GPT (IU/L)	25.8 ± 24.6	33.9 ± 24.9	<0.001
GGT (IU/L)	36.0 ± 42.7	55.3 ± 63.8	<0.001
Creatinine (mg/dL)	0.9 ± 0.2	1.0 ± 0.2	<0.001
Triglyceride (mg/dL)	108.0 ± 69.9	144.8 ± 95.6	<0.001
HDL cholesterol (mg/dL)	53.3 ± 12.6	49.3 ± 11.1	<0.001
LDL cholesterol (mg/dL)	121.8 ± 28.9	129.4 ± 31.1	<0.001
Urine albumin, Positive (*N*, %)	363 (1.2%)	203 (2.6%)	<0.001
Smoking *(N*, %)			<0.001
None	14,274 (47.1%)	2198 (28.5%)	
Ex-smoker	9891 (32.6%)	3375 (43.8%)	
Current smoker	6131 (20.2%)	2132 (27.7%)	
Alcohol, Heavy (*N*, %)	16,236 (53.6%)	5298 (68.8%)	<0.001
Diabetes, Yes (*N*, %)	2311 (7.6%)	508 (6.6%)	0.002
Hypertension, Yes (*N*, %)	6003 (19.8%)	2169 (28.2%)	<0.001
Dyslipidemia, Yes (*N*, %)	4765 (15.7%)	1531 (19.9%)	<0.001

**Table 4 jcm-08-00172-t004:** Comparison of model performance for maximum sensitivity criterion and maximum BCR criterion.

Model	Training Set						Test Set					
	Accuracy	SN	SP	BCR	Precision	F1 Score	Accuracy	SN	SP	BCR	Precision	F1 Score
For maximum sensitivity criterion												
DAC	0.70	0.58	0.73	0.65	0.35	0.44	0.70	0.59	0.73	0.65	0.37	0.45
KNNC	1	1	1	1	1	1	0.72	0.34	0.82	0.53	0.34	0.34
NBC	0.62	0.73	0.60	0.66	0.31	0.44	0.63	0.73	0.60	0.66	0.33	0.45
SVMC	0.53	0.48	0.54	0.51	0.21	0.29	0.52	0.48	0.54	0.51	0.22	0.30
DTC	0.80	0.10	0.97	0.31	0.52	0.17	0.78	0.08	0.97	0.28	0.49	0.14
RFC	0.78	0.88	0.75	0.81	0.47	0.61	0.68	0.66	0.69	0.67	0.36	0.47
For maximum BCR criterion												
DAC	0.70	0.58	0.73	0.65	0.35	0.44	0.70	0.59	0.73	0.65	0.37	0.45
KNNC	1.00	1.00	1.00	1.00	1.00	1.00	0.72	0.34	0.82	0.53	0.34	0.34
NBC	0.62	0.73	0.60	0.66	0.31	0.44	0.63	0.73	0.60	0.66	0.33	0.45
SVMC	0.53	0.48	0.54	0.51	0.21	0.29	0.52	0.48	0.54	0.51	0.22	0.30
DTC	0.80	0.10	0.97	0.31	0.52	0.17	0.78	0.08	0.97	0.28	0.49	0.14
RFC	0.73	0.71	0.73	0.72	0.40	0.51	0.70	0.64	0.71	0.68	0.37	0.47

SN: sensitivity; SP: specificity; BCR: balanced classification rate; DAC: discriminant analysis classification; KNNC: K-nearest neighbor classification; NBC: naïve Bayes classification; SVMC: support vector machine classification; DTC: decision tree classification; and RFC: random forest classification.

**Table 5 jcm-08-00172-t005:** Performance comparison with conventional logistic regression model for total set (maximum sensitivity criterion).

	AUC	95% Confidence Interval	*p* for Comparison with CLR
CLR	0.568	0.563–0.572	Reference
NBC	0.669	0.663–0.675	<0.001
RFC	0.775	0.770–0.780	<0.001
DAC	0.661	0.655–0.667	<0.001
KNNC	0.8723	0.868–0.877	<0.001
SVMC	0.515	0.509–0.522	<0.001
DTC	0.537	0.534–0.541	<0.001

CLR: conventional logistic regression; NBC: naïve Bayes classification; RFC: random forest classification; DAC: discriminant analysis classification; KNNC: K-nearest neighbor classification; SVMC: support vector machine classification; DTC: decision tree classification; and AUC: area under the curve.

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
