# Peer review of "Exploration of Machine Learning for Hyperuricemia Prediction Models Based on Basic Health Checkup Tests"

_jcm, 2019, doi:10.3390/jcm8020172_

Round 1
Reviewer 1 Report
The main goal of this article is the prediction of uric acid level (hyperuricemia) based on clinical data obtained through medical check-ups. This prediction may be important to prevent the occurrence of various chronic diseases caused by elevated uric acid levels.
In the first objective of trying to obtain a maximum value of sensitivity in the test set, it was verified that the two algorithms that had better performance values were Naïve Bayes and Random Forest, respectively. On the other hand, in the search of the models that had the highest value of BCR in the training set, the Random Forest classification algorithm, followed by Naïve Bayes, was highlighted. As regards the comparison of the performance of these algorithms with a more traditional Logistic Regression algorithm, it was found that the most sophisticated Machine Learning algorithms presented better AUC values. It can then be concluded that this type of algorithm has a greater predictive capacity. One of the reasons for this result is the exclusion of attributes that at the outset are not related to hyperuricemia, but which may be useful in the prediction process, by the traditional model.
Authors could have provided more information on the traditional method of Logistic Regression. They did not properly justify the choice of Machine Learning algorithms chosen for the study.
Author Response
1. Authors could have provided more information on the traditional method of Logistic Regression. They did not properly justify the choice of Machine Learning algorithms chosen for the study.
[Response] Thank you for the reviewer’s comment. As the reviewer recommended, we added some description to make the meaning more clear as below.
[Line 176-186] There are several reasons to choose ML algorithms over conventional statistical method for designing a prediction model. First, compared to conventional statistical analysis, ML can design a prediction model that reflects the relationship between variables without prior knowledge of the algorithm [23]. This characteristic makes it possible to include all information from the input data regardless of its effectiveness during analysis and prevents overseeing data with indefinite effectiveness. Second, in conventional statistical analysis, it is assumed that the input variables are independent [3]. However, this assumption is impossible in the real world. Various input factors are inter-related in complex ways, regardless of whether these ways are known or not. ML considers potential interactions so that all information in the input data can be reflected in the analysis [24], and it can improve prediction performance with complex, heterogenous, high-dimensional data [25].
Reviewer 2 Report
This is an interesting paper comparing a range of different ML techniques to a classification problem in a relative healthy population. The sample size is large and the methodology appears to be robust. The study is observational and highlights the value of the ML techniques. The discussion is extensive and valuable as is the set of references.
There are a few minor editorial issues.
line 49 - 'electrical medical record' should be 'electronic medical record'
line 52 - 'medical clinical' should be 'clinical'
line 55 - 'helpful for preventing' should be 'helpful in preventing'
line 66,67 - cholesterol is mentioned twice in the list
line 71 - uric acid level for males should be '70 mg/L' not '7.0'
Use 'p' rather than 'P' for the probability
In step 2 (line 82) clarify with an example how this averaging occured.
At what stage was the uric acid level actually measured, it is not stated in the text. I presume it was as part of the checkup but not reported externally.
In Table 2 something has happened with the formatting of the second column, 6 and 7 rows (Balanced classification rate and F1-score
In Table 3. What does the p value in row one (Sex, Male (N,%)) refer to? males with hyperuricemia, males?
The data in Table 3 suggests that the hyperuricemic group were not statistically different to the normal urate level cohort for every measure. This doesnt seem correct for the first row 65% versus 88%?
In table 5 the abbreviation KNNC is used whereas it was KNN previously.
Author Response
1. Speeling and formal errors such as
line 49 - 'electrical medical record' should be 'electronic medical record'
line 52 - 'medical clinical' should be 'clinical'
line 55 - 'helpful for preventing' should be 'helpful in preventing'
line 66,67 - cholesterol is mentioned twice in the list
Use 'p' rather than 'P' for the probability
[Response] Thank you for the reviewer’s considerate reviews. As the reviewer recommended, we corrected all the errors as recommended in the manuscript.
2. line 71 - uric acid level for males should be '70 mg/L' not '7.0'
[Response] Thank you for the reviewer’s comment on my big mistake. The correct level will be 7.0 mg/dL for men and 6.0 mg/dL for women, I had mistaken the number for women, and corrected in the manuscript
3. In step 2 (line 82) clarify with an example how this averaging occured.
[Response] Thank you for the reviewer’s comment. We’ changed the description to make the meaning more clear as below
[Line 82] we replaced these items with the mean value of the non-missing observations for the predictor.
4. At what stage was the uric acid level actually measured, it is not stated in the text. I presume it was as part of the checkup but not reported externally.
[Response] Thank you for the reviewer’s comment. We’ added some description to make the meaning more clear as below
[Line 61-62] Uric acid level was measured during comprehensive health checkups at Gangnam Center in addition to tests performed in national checkups.
5. In Table 2 something has happened with the formatting of the second column, 6 and 7 rows (Balanced classification rate and F1-score
[Response] Thank you for the reviewer’s comment. There might have been some technical errors. We’ve corrected it in the manuscript
6. In Table 3. What does the p value in row one (Sex, Male (N,%)) refer to? males with hyperuricemia, males?
[Response] we’ve separated the male’s and female’s statistics to make the meaning clear in Table 3.
7. The data in Table 3 suggests that the hyperuricemic group were not statistically different to the normal urate level cohort for every measure. This doesnt seem correct for the first row 65% versus 88%?
[Response] As you mentioned, all of the demographic feature is table 3 showed statistical diefference between normal level and hyperuricemic
group significantly, since the p-values were all less than 0.05.
8. In table 5 the abbreviation KNNC is used whereas it was KNN previously.
[Response] I unified the expression as KNNC throughout the manuscript.